# The Effect of Fertilization on Floristic Composition and Biodiversity of Montane Grasslands (HNV) in the Eastern Carpathians

**DOI:** 10.3390/plants15010080

**Published:** 2025-12-26

**Authors:** Emilian Canișag, Costel Samuil, Culiță Sîrbu, Adrian-Ilie Nazare, Bogdan-Ioan Grigoraş, Vasile Vîntu

**Affiliations:** Plant Science Department, Iasi University of Life Sciences, 3, M. Sadoveanu Alley, 700490 Iasi, Romania; emilian.canisag@iuls.ro (E.C.); costel.samuil@iuls.ro (C.S.); culita.sirbu@iuls.ro (C.S.); adrian.nazare@iuls.ro (A.-I.N.); bogdan.grigoras@iuls.ro (B.-I.G.)

**Keywords:** montane grasslands, High Nature Value (HNV), fertilization gradient, biodiversity, indicator species, Eastern Carpathians, sustainable grassland management

## Abstract

High Nature Value (HNV) mountain grasslands in the Eastern Carpathians are highly sensitive to changes in management intensity, particularly fertilization. This study assessed the effects of contrasting organic and mineral fertilization regimes on floristic composition, vegetation types, and diversity in an oligotrophic *Nardus stricta* grassland within an experimental framework established in 2021. The analysis is based on vegetation data collected over three consecutive years (2022–2024) from nine treatments, including an unfertilized control, organic fertilization with manure (10–30 t ha^−1^ applied in autumn or spring), and mineral fertilization with nitrocalcar (Nitrocalc_20—200 kg ha^−1^ calcium ammonium nitrate and Nitrocalc_30—300 kg ha^−1^ calcium ammonium nitrate). Vegetation responses were evaluated using hierarchical cluster analysis, principal coordinates analysis (PCoA), multi-response permutation procedures (MRPP), indicator species analysis (ISA), and α-diversity indices. Six floristic types were identified along a pronounced trophic gradient ranging from oligotrophic to eutrophic communities. Low to moderate organic fertilization (10–20 t ha^−1^) maximized species richness, diversity, and community evenness, maintaining a stable assemblage of oligotrophic and mesotrophic species. In contrast, high manure inputs (30 t ha^−1^) and mineral fertilization resulted in rapid floristic simplification, loss of oligotrophic indicators, and dominance of competitive grasses. These results indicate that moderate organic fertilization represents an effective adaptive management option for conserving HNV mountain grasslands, whereas intensive mineral fertilization is incompatible with biodiversity conservation objectives.

## 1. Introduction

Mountain grasslands are among the ecosystems most strongly affected by biodiversity loss under global change, primarily due to increasing anthropogenic pressures such as land-use intensification, nutrient enrichment, and abandonment. In regions where species-rich grasslands have historically been maintained through low-input traditional management, even moderate changes in fertilization intensity can trigger disproportionate ecological responses, leading to shifts in plant competition, vegetation structure, and ecosystem functioning.

High Nature Value (HNV) grasslands represent some of the most complex and valuable agro-pastoral ecosystems in Europe, characterized by high floristic richness, structural heterogeneity, and a long history of extensive management through mowing and grazing [1,2,3]. These systems provide a wide range of ecosystem services, including the production of high-quality fodder [4,5,6], maintenance of soil structure and nutrient cycling [7,8], carbon sequestration and hydrological regulation [9,10,11], habitat provision for biodiversity [12,13,14,15], and cultural services supporting mountain communities and traditional landscapes [16,17,18].

In Romania, extensive areas of HNV grasslands occur across the Eastern, Southern, and Western Carpathians, where they form biodiversity hotspots integrated into the Natura 2000 network. Particularly noteworthy are the *Nardus stricta*–dominated grasslands of the Eastern Carpathians, which represent unique oligotrophic ecosystems with high conservation value and exceptional sensitivity to anthropogenic disturbance [9,19,20]. These semi-natural grasslands host numerous species of conservation and economic interest, including medicinal plants such as *Arnica montana* [21,22,23], and play a central role in shaping traditional cultural landscapes [24,25]. Due to their sensitivity to management changes, mountain grasslands are widely regarded as indicators of socio-ecological resilience [23,26,27,28].

Over recent decades, socio-economic transformations have driven two opposing but equally problematic trends in mountain regions: agricultural intensification and land abandonment [29,30]. Fertilization represents one of the main drivers of vegetation dynamics in mountain grasslands, as it can enhance productivity and forage quality [31,32,33], but excessive inputs often lead to simplified vegetation structure, dominance of nitrophilic species, and biodiversity loss [34]. Conversely, abandonment promotes secondary succession, shrub encroachment, and declining forage value [35,36,37,38].

The ecological effects of fertilization depend strongly on dose, frequency, and nutrient source [35,39,40]. Low to moderate organic inputs can enhance productivity while maintaining high species diversity, consistent with the intermediate disturbance hypothesis [23,41,42,43,44]. In addition, the continued application of extensive management practices plays a key role in maintaining HNV grasslands, preserving traditional mountain landscapes, preventing land abandonment, and limiting physical soil and sward disturbances [45,46,47]. In contrast, repeated mineral fertilization favors competitive grasses, reduces oligotrophic species, and simplifies phytocenoses [48,49], with cascading effects on soil microbial processes and nutrient cycling [50,51].

At the European level, a growing body of evidence demonstrates that persistent fertilization in alpine and subalpine grasslands leads to a decline in species diversity and promotes the dominance of competitive grasses [52,53]. Conversely, numerous studies highlight that the maintenance of traditional, low-intensity management is essential for preserving species-rich grassland habitats and their associated ecosystem functions [11,27,54]. In the Carpathian region, where climatic, demographic, and socio-economic pressures are intensifying, understanding the interactions between fertilization regimes and vegetation structure has become a key prerequisite for developing sustainable management strategies aimed at conserving biodiversity and safeguarding mountain cultural landscapes [55,56,57,58,59].

Despite substantial advances in grassland ecology, experimental evidence addressing the specific effects of contrasting organic and mineral fertilization regimes on *Nardus stricta*–dominated HNV grasslands in the Eastern Carpathians remains limited. Most previous studies focus on broader geographic scales, short-term observations, or do not directly compare fertilizer doses and application timing within a single experimental framework. Addressing this gap is essential for understanding how fine-scale management differences shape floristic composition and biodiversity and for developing adaptive strategies that reconcile conservation objectives with agricultural production in sensitive mountain environments.

We hypothesized that (1) moderate organic fertilization would maintain or increase floristic diversity relative to unfertilized controls, in line with the intermediate disturbance hypothesis; (2) high organic and mineral fertilization would reduce species richness and promote competitive, eutrophic species; and (3) the timing of organic fertilizer application (autumn vs. spring) would differentially influence floristic composition and diversity. Accordingly, this study aimed to assess how different fertilization regimes (organic and mineral, moderate and intensive) affect floristic composition, vegetation types, and biodiversity in mountain grasslands of the Eastern Carpathians using an integrated analytical framework combining vegetation classification, PCoA, MRPP, and indicator species analysis. The study was conducted within an experimental framework established in 2021 and focuses on short-term vegetation responses (2022–2024), which provide insight into early-stage community dynamics and expected long-term trajectories under contrasting management intensities.

## 2. Results

### 2.1. Hierarchical Cluster Analysis of Floristic Composition

The hierarchical classification analysis of floristic composition revealed clear differences between the experimental variants, driven by fertilization intensity and type. Using a cutoff of approximately 72% of the remaining information allowed the delimitation of six distinct floristic groups, each corresponding to a specific ecological grassland type (Figure 1). Cluster 1 (control variant T1) was characterized by the dominance of acidophilic species, particularly *Nardus stricta* associated with *Festuca rubra*, reflecting an oligotrophic floristic composition. Cluster 2 included variants moderately fertilized with small doses of manure (T2 and T5). Reduced organic fertilization led to a decrease in the abundance of *Nardus stricta* and an increase in the proportion of *Festuca rubra*, associated with grasslands characterized by a moderate trophic level. Cluster 3 included variants at an advanced stage of the improvement process (T3 and T6). *Festuca rubra* and *Agrostis capillaris* dominated this group, but acidophilic elements were still present, positioned between oligotrophic and mesotrophic grassland types. Cluster 4 corresponded to a stable type of improved grassland dominated by *Festuca rubra* and *Agrostis capillaris*. This group is associated with moderate to high doses of manure applied in both autumn and spring, characterized by the co-dominance of *Festuca rubra* and *Agrostis capillaris* and a higher representation of mesotrophic species. Cluster 5 was characterized by the pronounced dominance of *Agrostis capillaris*, followed by *Festuca rubra*, especially under high doses of manure (T4 and T7). Vegetation was characterized by a pronounced dominance of *Agrostis capillaris*, followed by *Festuca rubra*. Cluster 6 includes variants intensively fertilized with nitrocalcar (T8, T9), where *Agrostis capillaris* becomes almost exclusively dominant. High mineral fertilization significantly reduces the presence of oligotrophic species and leads to simplification of the floristic structure. Overall, the dendrogram indicated clear differences between the control, organic fertilization, and mineral fertilization, reflecting a consistent gradient in floristic composition across fertilization treatments. The obtained groups formed the basis for the detailed ecological interpretations presented in the Discussion section.

### 2.2. Principal Coordinates Analysis (PCoA) and Floristic Differentiation

PCoA showed the clear separation of treatments according to the vegetation types previously identified by clustering analysis. Axis 1 explains 92.1% of the variation in species composition and represent the main fertilization gradient. Axis 2 contributed an additional 6.7%, primarily distinguishing unfertilized plots from those receiving organic fertilization applied in autumn (Table 1).

The strongest link was between Nitrocalc_30 and Axis 1 (r = 0.582, *p* < 0.01), indicating their position at the high-input end of the fertilization gradient. On Axis 2, only two treatments showed significant correlations: no fertilization (positive) and 20 t ha^−1^ manure in autumn (negative), highlighting the differentiated effect of the timing of manure application on vegetation structure. Overall, Axis 1 represented the oligotrophic–eutrophic gradient, and Axis 2 differentiated the moderate treatments according to the season of manure application (Table 2).

The PCoA (Figure 2) showed a clear separation of areas based on floristic composition, with the first two axes explaining almost all of the variation between treatments. The areas were organized into three large groups corresponding to the types dominated by *Nardus stricta*, *Festuca rubra*, and *Agrostis capillaris*. The unfertilized areas (T1) were grouped in the area associated with *Nardus stricta*, whereas the treatments with moderate levels of manure (T2–T6) were located in the intermediate area, which was characterized by the presence of *Festuca rubra*. Mineral treatments (T8–T9) and some variants with high doses of manure (T4 and T7) were located/grouped in the ordination space associated with *Agrostis capillaris* dominance. The overlaid vectors indicate the overall direction of the fertilization gradient. *No_fertilz* was oriented toward the *Nardus stricta* group. The vectors corresponding to high fertilization levels (*Nitrocalc_20*, *Nitrocalc_30*, *Fert_high_aut, Fert_high_spr*) pointed towards the *Agrostis capillaris* group. The contours of vegetation types (vegetation ‘*hilltops*’, i.e., areas of maximum density in ordination space) highlighted a clear spatial separation of the three main groups, with moderate overlap in the intermediate mesotrophic zone and with minimal overlap at the extremes of the gradient. The treatment gradient followed the expected transition from the unfertilized control (T1) on the left side of the ordination to the high nitrocalcar doses (T9) on the right side, reflecting progressive changes in floristic composition.

The relationships between the three dominant species (*Agrostis capillaris, Festuca rubra*, and *Nardus stricta*) and the PCoA axes confirmed their essential roles in defining trophic gradients (Figure 3). *Agrostis capillaris* (panel A) appeared in the positive area of Axis 1, coinciding with the positioning of treatments T8 and T9 and variants with high doses of manure (T4 and T7). The clear positive relationship between Axis 1 scores and species abundance indicates a positive association with higher fertilization levels. *Festuca rubra* (panel B). *Festuca rubra* occupied intermediate positions on Axis 1 and values close to the origin on Axis 2. Areas T2–T6 are in the vicinity of this species, reflecting its preference for moderate fertilization levels. The negative slopes of the regressions on both axes indicate that the abundance of *Festuca rubra* decreases towards both the oligotrophic and eutrophic extremes. *Nardus stricta* (panel C). *Nardus stricta* is positioned in the negative area of Axis 1 and in the positive part of Axis 2, in line with the grouping of control areas (T1). The negative relationships with Axis 1 and slightly positive relationships with Axis 2 highlight the preference of this species for oligotrophic soils with no or minimal fertilization. Overall, the overlap of treatment vectors over site scores and species abundance clearly demonstrated how the applied inputs shaped the floristic-ecological distribution of the communities.

The two main axes delimit four major floristic types (Figure 4A,B): Subtype *Nardus stricta—Festuca rubra*; Type *Festuca rubra—Nardus stricta*; Type *Festuca rubra—Agrostis capillaris*; Type *Agrostis capillaris*. These were coherently arranged in the ordination space. Unfertilized areas (T1) were concentrated in the *Nardus stricta* zone, whereas low and medium manure treatments (T2–T6) overlapped in the mesotrophic zone dominated by *Festuca rubra*. Mineral (T8–T9) and high manure (T4, T7) treatments were grouped in the eutrophic zone dominated by *Agrostis capillaris*. The treatment vectors reflect the direction of the trophic gradient: *No_fertilz* → oligotrophic zone; *Fert_high_aut/Fert_high_spr/Nitrocalc_20/Nitrocalc_30* → Eutrophic zone. The distribution of dominant species is consistent with this structure, confirming the concordance between classification, ordination, and ecologically defined floristic type. These floristic types, visually highlighted in the ordination space, were subsequently statistically validated using the MRPP analysis.

### 2.3. Multi-Response Permutation Procedure Analysis

MRPP analysis showed the strong statistical separation of the six floristic groups (Table 3). All pairwise comparisons were significant (*p* < 0.001) with high A values (0.284–0.767), indicating internal homogeneity above the expected random level. The highest A values were observed in comparisons involving the unfertilized control (Group 1) and the mineral fertilization treatments (Group 6), reflecting high within-group agreement. Group 6 (T8–T9) showed the highest internal homogeneity (A up to 0.767), whereas intermediate groups exhibited lower but still significant A values. Comparisons between Group 5 (high manure input) and Group 6 (mineral fertilization) also showed strong separation, indicating distinct floristic composition between intensive organic and mineral fertilization treatments. Thus, the MRPP validated the stability of the floristic types delineated by the cluster and PCoA.

### 2.4. Species Distribution Along PCoA Ordination Axes

The distribution of species in relation to the ordination axes showed distinct patterns of association along the fertilization gradient (Table 4). On Axis 1, positive values were mainly associated with variants fertilized with nitrocalcar (T8–T9) and variants with high doses of manure (T4–T7). Species with the highest positive correlations included *Agrostis capillaris* and *Phleum pratense*. Negative values on Axis 1 were associated with unfertilized areas (T1) and treatments with low doses of manure applied in autumn and spring (T2 and T5, respectively). This part of the ordination space included species such as *Nardus stricta*, *Festuca rubra*, *Scorzonera rosea*, *Potentilla erecta*, and other species characteristic of areas with low inputs. The association of these species with the negative area of Axis 1 corresponds to treatments in which the influence of fertilization was reduced or absent. On Axis 2, the differentiation was observed between unfertilized areas, variants with manure applied in autumn, and treatments with medium-to high organic inputs. Species such as *Nardus stricta*, *Viola canina*, and *Luzula luzuloides* were positioned toward positive values of Axis 2, whereas species with negative correlations included *Trifolium repens*, *Anthoxanthum odoratum*, *Veronica chamaedrys*, *Lotus corniculatus*, and *Trifolium pratense*, were associated with variants with medium organic inputs and transitions between moderately and intensively fertilized areas. These species were found mainly in the space occupied by treatments T3, T4, and T6, where the effect of manure, regardless of the time of application, was reflected in the floristic composition.

### 2.5. Indicator Species Analysis (ISA)

Indicator species analysis (ISA) identified distinct and ecologically meaningful sets of characteristic species for each of the six floristic groups defined by the hierarchical cluster analysis (Table 5). Each group was associated with at least one species exhibiting a high Indicator Value (IndVal) and statistical significance, confirming the robustness of the floristic typology derived from the fertilization gradient.

Group 1, corresponding to oligotrophic grasslands dominated by *Nardus stricta*–*Festuca rubra*, showed the most consistent assemblage of indicator species, including *Nardus stricta*, *Avenula planiculmis*, *Potentilla erecta*, *Viola canina*, and *Luzula luzuloides*. Group 2 (communities with low manure inputs, 10 t ha^−1^) was mainly associated with *Phleum alpinum*, *Poa chaixii*, *Rhinanthus minor*, and *Hypochaeris uniflora*. Group 3 (communities under moderate spring manure inputs) was characterized by species that prefer moderate trophic input, such as *Anthoxanthum odoratum*, *Trifolium hybridum*, *Trifolium repens*, *Lotus corniculatus*, *Hypericum maculatum*, *Leucanthemum vulgare*, *Cruciata glabra*, and *Veronica chamaedrys*. Group 4 (communities with moderate autumn manure inputs) was defined by the presence of *Trifolium pratense* and *Centaurea Phrygia*. Group 5 (communities with high manure inputs) showed high indicator values for *Alchemilla xanthochlora* and *Dianthus barbatus* subsp. *compactus*, species characteristic of high doses of manure applied in autumn and spring, respectively. Group 6 (communities with mineral fertilization) was strongly associated with clear indicators of mineral fertilization, such as *Agrostis capillaris* and *Phleum pratense*. Both displaying high IndVal values and an almost exclusive occurrence in the nitrocalcar variants. Overall, the ISA clearly highlighted a shift from stress-tolerant oligotrophic indicators (Group 1) to competitive, fast-growing grasses under mineral fertilization (Group 6), providing strong ecological support for the fertilization gradient identified by cluster analysis and PCoA ordination. ISA validated the robustness of the identified floristic types, confirming the response of plant communities to fertilization gradients.

### 2.6. The Impact of Management Scenarios on Diversity Indices

The diversity index showed a clear, non-linear response of grassland communities to the fertilization gradient (Table 6).

The unfertilized control plots (T1) exhibited moderate values of species richness and diversity (S = 31.75; H′ = 2.199), characteristic of oligotrophic mountain grasslands with relatively balanced species abundances.

Treatments receiving low organic inputs (T2 and T5; 10 t ha^−1^ manure) displayed higher diversity values compared to the control. Among these, treatment T5 (spring application) showed the highest species richness and Shannon diversity (S = 35.00; H′ = 2.857).

Moderate manure inputs (T3 and T6; 20 t ha^−1^) resulted in consistently high diversity and evenness values (H′ ≈ 2.70–2.77; E ≈ 0.78–0.80), indicating a relatively uniform distribution of species abundances across these treatments.

High organic fertilization (T4 and T7; 30 t ha^−1^) was associated with a decline in diversity and evenness, particularly in T7 (H′ = 1.958; S = 24.25), reflecting increased dominance within the plant community.

The lowest diversity values were recorded under mineral fertilization treatments (T8 and T9), with T9 showing minimal species richness and diversity (S = 11.25; H′ = 0.926), together with the lowest evenness and Simpson index values.

One-way ANOVA indicated significant differences among treatments for all diversity indices (species richness, Shannon index, evenness, and Simpson index; *p* < 0.001), confirming a strong and consistent effect of fertilization intensity and type on grassland biodiversity.

## 3. Discussion

### 3.1. Floristic Responses to Fertilization in HNV Mountain Grasslands

The results of hierarchical classification and PCoA ordination clearly show that the fertilization gradient profoundly alters the structure of plant communities, generating clear transitions between three major vegetation types: *Nardus stricta* (oligotrophic), *Festuca rubra* (mesotrophic), and *Agrostis capillaris* (eutrophic). The evolution of these floristic types is in line with the patterns reported in HNV grasslands in Romania [14,60,61] and Central Europe, where excessive fertilization leads to the simplification of vegetation cover and loss of specialized species [62,63]. In the Lucina experiment, low–moderate organic fertilization favored the *Festuca rubra—Nardus stricta* and *Festuca rubra—Agrostis capillaris* types, confirming that moderate nutrient levels can improve trophic conditions without compromising diversity. Similar results were reported in the Apuseni Mountains and Transylvania, where moderate doses of manure maintained a stable core of mesotrophic and oligotrophic species [23,45,64]. In contrast, intensive organic and mineral fertilization led to the almost exclusive dominance of *Agrostis capillaris*, indicating that the resilience threshold of HNV habitats had been exceeded. Similar results regarding floristic simplification under intensive fertilization have been reported in mountain grasslands in Romania [65] and in long-term experiments in Central Europe [62,66]. These studies show that once fertilization exceeds moderate levels, semi-natural communities tend to be dominated by a few competitive grasses, with the loss of oligotrophic species of conservation value and HNV indicators [63,67]. The exceptionally high proportion of variation explained by Axis 1 (92.1%) indicates that fertilization intensity represents the dominant ecological filter controlling plant community composition in the studied grasslands. This result suggests that nutrient input, particularly nitrogen availability, overwhelmingly structures vegetation patterns, largely overriding other potential drivers such as microtopographic variability, background soil heterogeneity, or minor climatic differences within the experimental area. The strong unidirectional gradient captured by Axis 1 reflects a rapid shift from stress-tolerant oligotrophic communities dominated by *Nardus stricta* to competitive, fast-growing grass assemblages dominated *by Agrostis capillaris*. Such dominance of a single ordination axis has been reported in other fertilization experiments and is characteristic of systems where nutrient enrichment acts as the principal determinant of species turnover. This pattern is consistent with niche-based competition theory, whereby increasing nutrient availability reduces belowground resource limitation and shifts competitive interactions toward aboveground processes, particularly light competition, favoring fast-growing grasses with high biomass accumulation rates [62,63,67]. It should be emphasized that the patterns observed in the present study reflect early-stage vegetation responses to fertilization, as the experimental treatments were established recently. These initial shifts in dominance and species composition provide insight into the direction of community change under different nutrient regimes.

In the context of HNV, and as confirmed by our results, high and repeated organic and mineral fertilization is incompatible with the maintenance of oligotrophic types of *Nardus stricta*, as defined in the recent literature [55,68,69]. From this perspective, the results from the Eastern Carpathians converge with those reported in the Apuseni Mountains for *Arnica montana* grasslands, where moderate manure use has allowed the conservation of HNV habitats, whereas intensification has led to the loss of sensitive species [21,22,70]. Based on evidence from comparable long-term experiments in mountain grasslands, these early-stage responses are likely to represent the initial phase of longer-term successional trajectories, in which sustained high nutrient inputs progressively reinforce competitive exclusion, while moderate organic inputs may stabilize mesotrophic communities with high conservation value. In this context, the observed early dominance shifts can be interpreted as precursors of long-term alternative stable states driven by sustained differences in nutrient inputs, as documented in long-term grassland experiments across Europe [62,66]. The MRPP results further support this interpretation by demonstrating strong internal consistency within floristic groups and clear compositional thresholds between fertilization regimes. In particular, the high within-group agreement observed for unfertilized and mineral-fertilized plots indicates that these management extremes rapidly impose distinct and stable vegetation states, while intermediate organic treatments promote more gradual compositional shifts.

### 3.2. Species–Environment Relationships and Indicator Species Patterns

The correlations between species and the PCoA axes confirmed the role of dominant grasses as effective indicators of trophic gradient. *Nardus stricta* and the associated oligotrophic species (e.g., *Potentilla erecta*, *Luzula luzuloides*, *Avenula planiculmis*, *Euphrasia stricta*, *Viola canina*) are strongly associated with unfertilized areas, characterizing the oligotrophic type typical Natura 2000 priority habitat 6230, corresponding to species-rich *Nardus* grasslands on siliceous substrates in mountain areas as defined under Annex I of the EU Habitats Directive.

Nitrocalcar treatments were characterized almost exclusively by *Agrostis capillaris* and *Phleum pratense*, which had the highest *IndVal* values in group 6. This pattern is typical of eutrophicated habitats, where competition for light and nutrients favors the growth of fast-growing grasses [12,67]. ISA thus validated the six floristic types identified, confirming that fertilization is the determining factor of the vegetation structure. These patterns are consistent with international studies showing that in semi-natural grasslands, nitrogen fertilization rapidly reduces diversity and favors a small set of competitive grasses [71]. Recent meta-analyses have confirmed that mineral fertilization tends to homogenize communities and reduce the importance of eutrophication-sensitive species, whereas moderate organic fertilization can maintain a higher degree of functional heterogeneity [8,72,73,74].

This group of species largely corresponds to the sets of indicators used to define oligotrophic grasslands in the Apuseni Mountains [19,27,58], and other European mountain regions [63]. Many of these species have been proposed as indicators for the management of HNV grasslands [15]. Their consistent association with low-input treatments in this study further supports their use as sensitive bioindicators of nutrient enrichment thresholds in mountain HNV grasslands [19,27,58]. Mesotrophic species, especially *Festuca rubra*, *Trifolium repens*, *Anthoxanthum odoratum*, and *Lotus corniculatus*, showed intermediate correlations, being sensitive to both intensification and reduction in inputs. These marked the optimal biodiversity zones identified in treatments with moderate manure. Such responses have also been described in other grassland systems [13], where moderate organo-mineral inputs maintained a balance between forage value and the presence of oligotrophic species. Our results are consistent with those obtained in the Apuseni Mountains on *Festuca rubra* grasslands fertilized with manure [49].

### 3.3. Diversity Responses to Organic vs. Mineral Fertilization

The analysis of diversity indices showed a typical response to the “*intermediate*” curve—the maximum values of specific Richness, Shannon and Simpson appeared in treatments with low or moderate doses of manure, especially T5 and T3/T6, in both autumn and spring. These diversity patterns represent early-stage expressions of the intermediate disturbance hypothesis, indicating that moderate organic fertilization initially relaxes competitive pressure without triggering dominance by fast-growing grasses. Under prolonged fertilization, however, similar systems have been shown to shift beyond this optimum, with delayed dominance effects and further diversity losses becoming evident over longer timescales [62,71]. This structure corresponds to the intermediate disturbance hypothesis [42,43,66] and is consistent with the results obtained in other mountain experiments in Romania [49], as well as in Central Europe [62]. Moderate organic fertilization maintained a balanced distribution of species abundance, reflecting reduced competition between dominant grasses and oligotrophic species. Similar results have been reported in recent meta-analyses, which show that moderate organic amendments increase functional diversity and ecosystem stability [8,72,75,76]. In contrast, mineral fertilization with nitrocalcar reduced diversity to extremely low levels, confirming the trend towards uniformity and loss of sensitive species. This situation has been confirmed by studies on HNV grasslands in Romania [49] and international research highlighting the destabilizing effect of eutrophication on the productivity and stability of grassland ecosystems [71,77]. The results of the experiment confirm that there is an “*optimal window*” in which moderate organic fertilization simultaneously maximizes productivity and diversity, a trade-off supported by recent studies [73,78,79].

### 3.4. Management Implications for Nardus Stricta Grasslands in the Eastern Carpathians

From the perspective of HNV grassland management in the Eastern Carpathians, the results indicate three main findings. Maintaining oligotrophic types (*Nardus stricta*). The reference grasslands (T1) and those with very low fertilization represent the core of oligotrophic habitats 6230 and HNV, comparable to those described in studies conducted in the Apuseni Mountains and Natura 2000 sites [69]. These ecosystems are home to numerous indicator species and plants of conservation and economic value [45,46,47] and are sensitive to both intensification and abandonment [38,51]. They are a priority for conservation and agricultural environmental schemes.

Moderate organic fertilization as an adaptive management tool. Doses of 10–20 t ha^−1^ (autumn or spring) increased diversity and production without compromising community structure. This “*optimal window*” is applicable for both maintaining HNV habitats and sustainable fodder production in the future. Similar results have been obtained in the Apuseni Mountains and other HNV grasslands, where technologies such as mulching combined with organic fertilization have maintained high levels of diversity and a relatively stable community composition [79,80].

Limiting mineral fertilization and high doses of manure. Mineral fertilization (T8–T9) and high manure doses (T4, T7) greatly simplified the vegetation. Furthermore, the results regarding species structure and ISA values are compatible with the use of indicator species as a rapid diagnostic tool for the agrochemical status of the soil and for assessing management quality [64]. The integration of these indicators with modern monitoring methods, such as multivariate ordination [81,82,83] and remote sensing [55,84,85,86], can support the development of adaptive management schemes for HNV grasslands in the Carpathian region of Romania.

## 4. Materials and Methods

### 4.1. Study Area and Pedoclimatic Conditions

The experiment was conducted on a permanent *Nardus stricta* grassland in the Lucina area of the Lucina–Fundu Moldovei Depression of the major geomorphological unit Obcinele Bucovinei (Eastern Carpathians, northeastern Romania) in a mountainous region characterized by a cool and humid climate typical of the lower mountain range. The experimental field was located at an altitude of 1220 m, eastern longitude 25°10′44″ and northern latitude 47°38′44″, with a 10% slope and northern exposure. The soil is classified as a white skeletal clinogleic luvisol. In the 0–28 cm soil layer, it is characterized by a low degree of base saturation, ranging between 28 and 47%. The soil shows strong acidity, with pH values between 4.5 and 4.6, and a low total cation exchange capacity (6.7–13.9 meq 100 g^−1^ soil). It is poorly supplied with available phosphorus (10–15 mg kg^−1^) and very poorly supplied with potassium (4–9 mg kg^−1^). The carbon-to-nitrogen ratio (C/N = 11.3) indicates a relatively slow decomposition of organic matter. In addition, the soil is affected by excess stagnant moisture. The peat layer on the soil surface, approximately 5–7 cm thick, promotes faster infiltration of water from precipitation and prevents water loss through direct evaporation, influencing plant growth and development. The surface layer showed an accumulation of plant debris on the surface and low natural fertility, which is characteristic of oligotrophic grasslands. This study is based on a field experiment established in 2021 and focuses on early vegetation responses in mountain grasslands, as reflected by floristic composition, vegetation types, and biodiversity recorded during the 2022–2024 period. The average annual temperature in the area is approximately 6–7 °C, and the annual precipitation frequently exceeds 650–700 mm, with maximums recorded in the summer months (May–August) and minimums in winter. The rainfall and temperature regimes are typical of mountain depressions, with high atmospheric humidity throughout the year. Before the experiment was set up, the grassland was used extensively in a traditional system (mowing and/or grazing with low stocking rates) without systematic mineral or organic fertilization, with nutrients primarily coming from animal manure.

The multi-year average temperature is 6.4 °C, and the average temperature during the growing season is 12.7 °C. During the period 2022–2024, both annually and during the growing season, the average temperature was higher than the multi-year average by 0.5–1.3 °C (Appendix A) and 0.1–0.9 °C, respectively, and the bioactive period (with temperatures above 0°C) in the studied area was 255 days, between March 10 and November 20. The multi-annual average precipitation is 738.6 mm, with the following distribution by season: spring 175.1 mm (23.7%), summer 287.8 mm (39%), autumn 154.0 mm (20.8%), and winter 121.7 mm (16.5%). During the growing season (April–September), the multi-annual average precipitation is 493.0 mm, which represents 66.7% of the annual amount (Appendix A). During the period 2022–2024, the annual precipitation was 65 mm (9%) higher than the multi-year average, and during the growing season, it was close to the multi-year average. The highest amounts of precipitation were recorded in June and July, with positive deviations between 33.1 and 42.8 mm from the multi-year average. 

The studied area is characterized by an average annual number of days in the cold season with snow cover between 79 and 100 days, an annual potential evapotranspiration of 599 mm, of which 338 (56.4%) in June–August, and a duration of sunshine of 1600–1800 h/year, of which 1200–1300 h (60%) in the warm season (April–September) and 400-500 h/year (40%) in the cold season.

### 4.2. Experimental Design

A single-factor experiment was organized using the randomized block method, with nine treatment variants and four repetitions, installed on a permanent grassland derived from *Nardus stricta* L. (*Nardus stricta-Festuca rubra*). The experimental plots were located on gently sloping terrain under homogeneous site conditions to reduce the influence of soil and relief microvariability on the results of the study. The research covered four blocks, and the experiment had the following variants: T1—Unfertilized control; T2—10 t ha^−1^ well-decomposed horse manure applied in autumn; T3—20 t ha^−1^ well-decomposed horse manure applied in autumn; T4—30 t ha^−1^ well-decomposed horse manure applied in autumn; T5—10 t ha^−1^ well-decomposed horse manure applied in spring; T6—20 t ha^−1^ well-decomposed horse manure applied in spring; T7—30 t ha^−1^ well-decomposed horse manure applied in spring; T8—Nitrocalc_20—200 kg ha^−1^ calcium ammonium nitrate applied in spring; T9—Nitrocalc_30—300 kg ha^−1^ calcium ammonium nitrate applied in spring. The horse manure used in the experiment was well decomposed (more than two years old) and originated from a traditional husbandry system, providing a complex supply of nutrients and organic matter to the soil. Its chemical composition was as follows: pH 6.7, P_2_O_5_ 0.334%, K_2_O 0.691%, total nitrogen 0.434%, Na 0.0544%, Mg 0.1197%, Zn 5.59 mg/kg, Cu 1.358 mg/kg; Fe 65.06 mg/kg. Mineral fertilization was performed with nitrocalcar (defined here as calcium ammonium nitrate—CAN; 27% total N—approximately equal proportions of nitrate-N and ammonium-N) supplemented with calcium and magnesium carbonate (8–12% CaCO_3_ + MgCO_3_). The balanced proportion between the nitric and ammoniacal forms ensures both rapid availability and gradual release of N. This fertilizer provides readily available nitrogen while partially buffering soil acidification, making it suitable for application in mountain grasslands. Fertilization was applied annually in spring (T5–T9) and November for the autumn treatment (T2–T4). Mechanization was minimal; mowing was performed once a year (in July), and the biomass was removed. The entire perimeter was fenced to prevent accidental grazing by animals.

### 4.3. Studies of Floral Composition

Floristic composition was assessed annually (in July) using the Braun–Blanquet scale modified for mountain grasslands [49]. The classic 1–5 scale was subdivided into subclasses a–c to increase the accuracy of the coverage estimates, a modification that is particularly beneficial for capturing subtle changes in species abundance in diverse and often finely structured mountain grassland communities (Figure 5 and Table 7) [87]. Floristic surveys were conducted annually in early July, when most grasses were in the flowering stage, allowing for accurate species identification and adequate comparisons between years and treatments [85]. The coverage of each species was estimated in each experimental plot (T1–T9, with four replicates per plot). Taxonomic determination of species was carried out with the help of specialized floristic works and by consulting international databases (POWO—Plants of the World Online, Euro + Med PlantBase) to ensure the use of updated and uniform nomenclature. To reduce disruptive influences, the area was protected, and mowing was performed uniformly at the same height after the surveys.

### 4.4. Data Analysis

Multivariate analyses were performed using PC-ORD version 7 software, which is widely applied in community ecology for classification, ordination, and testing differences among groups [82,83]. Floristic–ecological classification was performed to group the plots into grassland types using hierarchical cluster analysis based on the Sørensen (Bray–Curtis) similarity index and the unweighted pair group method with arithmetic mean (UPGMA) linkage. This combination (Bray–Curtis + UPGMA) is considered a robust standard in plant community analysis and is frequently used to delineate phytosociological and ecological groups in semi-natural grasslands [88,89]. The dendrogram cutoff level was set at approximately 72% of the remaining information, which allowed the identification of six distinct floristic groups with clear ecological interpretations (types and subtypes of grasslands dominated by *Nardus stricta*, *Festuca rubra*, and *Agrostis capillaris*). To illustrate the trophic gradient generated by the fertilization treatments and to separate vegetation types, we performed a Principal Coordinate Analysis (PCoA) using the Bray–Curtis distance. PCoA provides a unique and reproducible solution and allows for the direct interpretation of distances among plots, which is an important advantage over iterative methods such as the NMDS [34,90]. The first two ordination axes explained over 98% of the floristic variation. The treatments were introduced into the prescription as vectors (“*joint plots*”), representing the input gradient: *No_fertilz*—unfertilized control (T1); *Fert_low_aut*—10 t ha^−1^ manure in autumn (T2); *Fert_med_aut*—20 t ha^−1^ manure in autumn (T3); *Fert_high_aut*—30 t ha^−1^ manure in autumn (T4); *Fert_low_spr*—10 t ha^−1^ manure in spring (T5); *Fert_med_spr*—20 t ha^−1^ manure in spring (T6); *Fert_high_spr*—30 t ha^−1^ manure in spring (T7); *Nitrocalc_20*–200 kg ha^−1^ nitrocalcar (T8); *Nitrocalc_30*–300 kg ha^−1^ of nitrocalcar (T9). The vectors were normalized, and their statistical significance was tested using permutation tests (n = 999).

Testing differences between groups. The statistical separation of floristic groups was tested using the Multi-Response Permutation Procedure (MRPP), which calculates the following metrics: T statistic (differentiation between groups), A coefficient (intragroup homogeneity), and *p*-value (statistical significance threshold of α = 0.05). All pairwise comparisons between groups were tested, and A and T values were used to support the robustness of the floristic types associated with the treatments [81]. To identify the species characteristics of each type of grassland, *Indicator Species Analysis* (ISA) was used, according to the method of [91]. For each species, IndVal values (combination of frequency and relative abundance in the group) were calculated, and statistical significance was tested using permutations (n = 999). Species with high IndVal and *p* < 0.05 were considered indicator species relevant to the floristic groups corresponding to the six clusters.

To complement the vegetation-based multivariate analyses described above, an integrative ecological perspective was adopted, acknowledging the underlying soil–plant interactions. Although fertilization-induced changes in plant communities are ultimately mediated by soil physicochemical and biological processes, the present study focuses on vegetation-based indicators (floristic composition, diversity indices, ordination, and indicator species) as integrative early-stage responses of the soil–plant system to contrasting fertilization regimes. This approach allows the detection of ecosystem-level responses while the experiment continues to develop toward long-term assessments of soil biological and biogeochemical feedbacks.

### 4.5. Diversity Indices and Statistical Analysis

For each treatment and repetition, the classic α-diversity indices were calculated as follows: Specific richness (S)—total number of species per plot; Shannon–Wiener index (H′), which combines species richness and relative abundance; Evenness (E), calculated as the ratio between the observed H′ and the maximum possible H′ for the number of species (E = H′/ln S); Simpson index (D), which is sensitive to the dominance of abundant species in the community.

These indices are widely used in grassland ecology studies and are considered robust tools for assessing plant community diversity and balance [71,75,86]. The formula and interpretation of these indices followed standard recommendations in community ecology [92] and are widely used in semi-natural grassland studies. The differences between treatments (T1–T9) for each index were tested using a one-way analysis of variance (ANOVA). The assumptions of normality and homogeneity of variance were checked before applying ANOVA [93,94], where conditions were met, and means were separated using the LSD test at *p* < 0.05. The values presented in the diversity table (Table 6) are means ± standard error (n = 4 replicates per treatment).

## 5. Conclusions

This study demonstrates that fertilization is a determining factor of the structure and diversity of *Nardus stricta* mountain grasslands in the Eastern Carpathians, Romania. The results showed a well-defined trophic gradient from oligotrophic communities rich in conservatively valuable species to increasingly simplified phytocenoses as the nutrient inputs increased. Moderate organic fertilization (10–20 t ha^−1^ manure), applied in autumn or spring, supported the highest floristic diversity, confirming the existence of an optimal management window in which productivity and biodiversity can be simultaneously maintained. These treatments favored the intermediate vegetation types (*Festuca rubra—Nardus stricta* and *Festuca rubra—Agrostis capillaris*), characterized by structural heterogeneity and a stable sets of indicator species.

In contrast, high doses of manure (30 t ha^−1^) and, in particular, mineral fertilization with nitrocalcar (200–300 kg ha^−1^) led to strong reductions in species richness, loss of oligotrophic species, and establishment of eutrophicated communities dominated by *Agrostis capillaris*. This shift is consistent with competitive exclusion mechanisms driven by increased nutrient availability and light monopolization and is incompatible with the conservation objectives of the Natura 2000 network and High Nature Value grassland management.

Although based on early-stage vegetation responses following the establishment of treatments, the observed patterns provide clear indications of expected long-term successional trajectories under contrasting fertilization regimes. Overall, the results support moderate organic fertilization as an adaptive management tool for sensitive mountain grasslands, while highlighting that mineral fertilization should be avoided in these ecosystems.

These responses are therefore interpreted as indicative of expected long-term successional trajectories, rather than as final equilibrium states of community composition [95,96,97].

### Study Limitations and Future Research Directions

Longer-term monitoring of this experimental system will be essential to validate whether the early-stage responses identified here translate into stable long-term trajectories or whether delayed soil–plant feedbacks modify community development over time. Further integration of data on the soil microbiome [50], edaphic fauna [12,77], and ecosystem services [17,98] would allow for a more comprehensive assessment of the effects of fertilization on the functioning of HNV ecosystems. Despite these limitations, the results clearly demonstrate that moderate organic fertilization can simultaneously support productivity and biodiversity, whereas mineral intensification leads to the loss of HNV habitats. Integrating vegetation-based indicators with soil biological and physicochemical data will be particularly important for disentangling early aboveground responses from delayed belowground feedbacks in fertilized HNV grasslands [99,100,101].

Finally, the absence of direct soil physicochemical and biological indicators represents a limitation of this study, emphasizing the need for future research integrating soil–plant feedbacks to better elucidate the mechanisms underlying vegetation responses to fertilization. Multivariate analyses combined with indicator species remain valuable tools for the rapid diagnosis and monitoring of grassland condition under different management scenarios.

## Figures and Tables

**Figure 1 plants-15-00080-f001:**
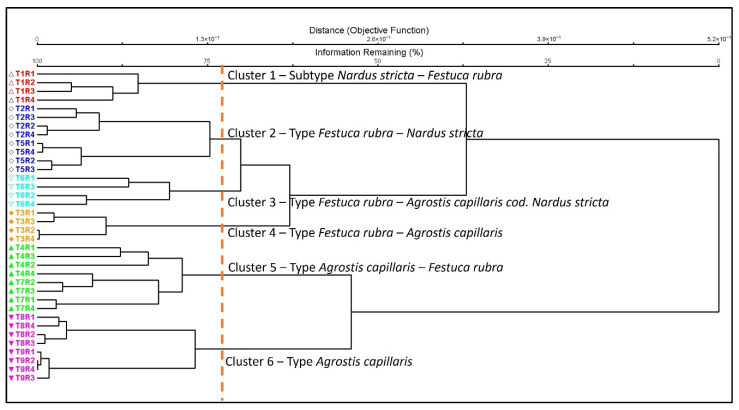
Floristic classification of the vegetation and changes in grassland type. Legend: T1—unfertilized control; T2—10 t ha^−1^ manure applied in autumn; T3—20 t ha^−1^ manure applied in autumn; T4—30 t ha^−1^ manure applied in autumn; T5—10 t ha^−1^ manure applied in spring; T6—20 t ha^−1^ manure applied in spring; T7—30 t ha^−1^ manure applied in spring; T8—Nitrocalc_20—200 kg ha^−1^ calcium ammonium nitrate applied in spring; T9—Nitrocalc_30—300 kg ha^−1^ calcium ammonium nitrate applied in spring; R1–R4—replications. Deshed line: the dendrogram cutoff line was set at approximately 72% of the remaining information, resulting in six clusters being identified.

**Figure 2 plants-15-00080-f002:**
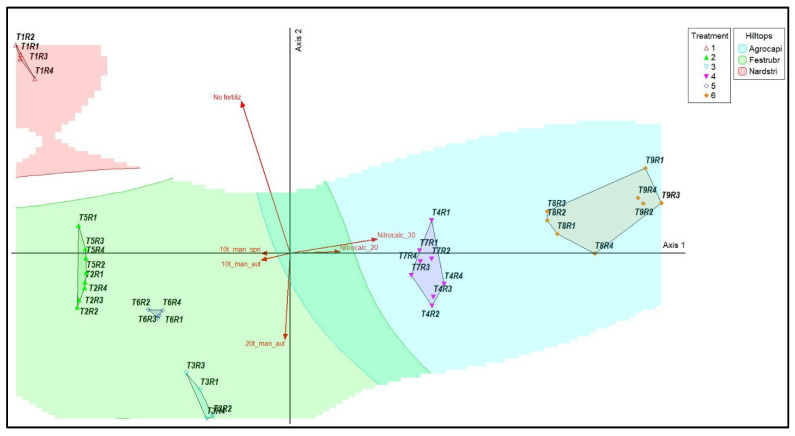
Principal Coordinates Analysis (PCoA) showing the floristic differentiation of mountain grasslands under manure and mineral fertilization treatments, with vegetation types delimited by the dominance of *Agrostis capillaris*, *Festuca rubra*, and *Nardus stricta*. Legend: T1—unfertilized control; T2—10 t ha^−1^ manure applied in autumn; T3—20 t ha^−1^ manure applied in autumn; T4—30 t ha^−1^ manure applied in autumn; T5—10 t ha^−1^ manure applied in spring; T6—20 t ha^−1^ manure applied in spring; T7—30 t ha^−1^ manure applied in spring; T8—Nitrocalc_20—200 kg ha^−1^ calcium ammonium nitrate applied in spring; T9—Nitrocalc_30—300 kg ha^−1^ calcium ammonium nitrate applied in spring; R1–R4—replications. Vectors indicate the fertilization gradient: *no _fertilz*, no fertilization (unfertilized control); *Fert_low_aut*, 10 t ha^−1^ manure applied in autumn (T2); *Fert_med_aut*, 20 t ha^−1^ manure applied in autumn (T3); *Fert_high_aut*, 30 t ha^−1^ manure applied in autumn (T4); *Fert_low_spr*, 10 t ha^−1^ manure applied in spring (T5); *Fert_med_spr*, 20 t ha^−1^ manure applied in spring (T6); *Fert_high_spr*, 30 t ha^−1^ manure applied in spring (T7); *Nitrocalc_20*, 200 kg ha^−1^ nitrocalcar applied in spring (T8); *Nitrocalc_30*, 300 kg ha^−1^ nitrocalcar applied in spring (T9).

**Figure 3 plants-15-00080-f003:**
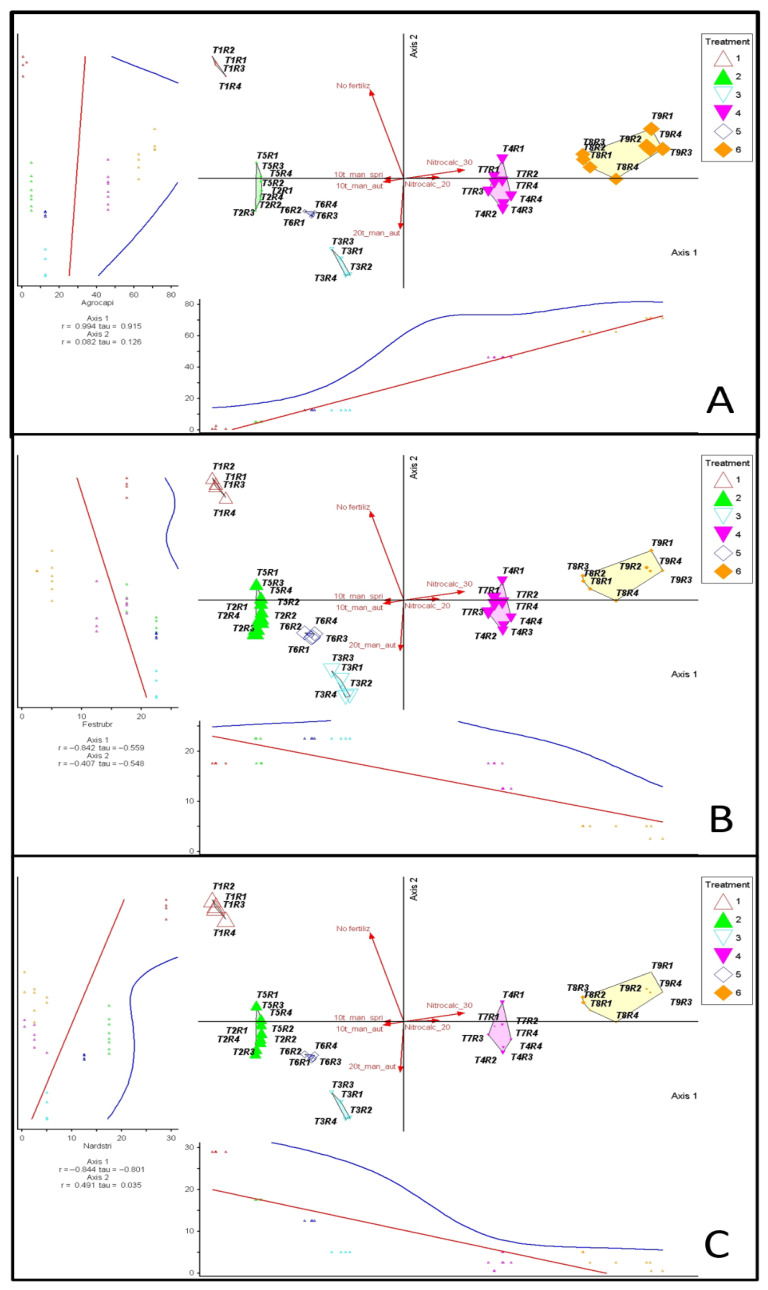
Species–environment relationships for the dominant grasses in the studied grasslands. Panels: (**A**) *Agrostis capillaris*, (**B**) *Festuca rubra*, (**C**) *Nardus stricta*. Each panel combines (i) principal coordinates analysis (PCoA) scores of vegetation plots, (ii) fitted environmental vectors (fertilization treatments), and (iii) species–environment correlations along the PCoA axes. Legend: T1—unfertilized control; T2—10 t ha^−1^ manure applied in autumn; T3—20 t ha^−1^ manure applied in autumn; T4—30 t ha^−1^ manure applied in autumn; T5—10 t ha^−1^ manure applied in spring; T6—20 t ha^−1^ manure applied in spring; T7—30 t ha^−1^ manure applied in spring; T8—Nitrocalc_20—200 kg ha^−1^ calcium ammonium nitrate applied in spring; T9—Nitrocalc_30—300 kg ha^−1^ calcium ammonium nitrate applied in spring; R1–R4—replications. Vectors indicate the fertilization gradient: *No_fertilz*—no fertilization (unfertilized control); *Fert_low_aut*—10 t ha^−1^ manure applied in autumn (T2); *Fert_med_aut*—20 t ha^−1^ manure applied in autumn (T3); *Fert_high_aut*—30 t ha^−1^ manure applied in autumn (T4); *Fert_low_spr*—10 t ha^−1^ manure applied in spring (T5); *Fert_med_spr*—20 t ha^−1^ manure applied in spring (T6).

**Figure 4 plants-15-00080-f004:**
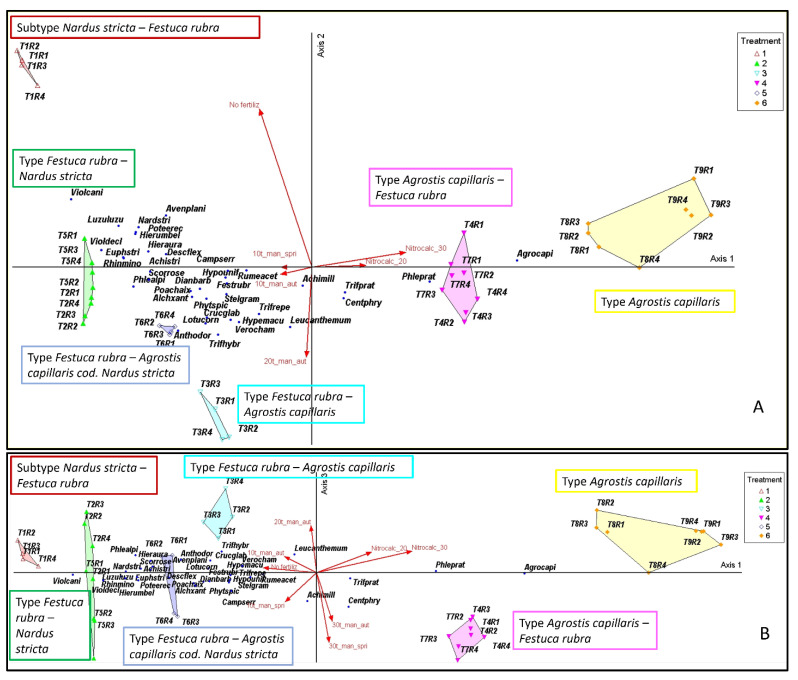
(**A**,**B**). Principal Coordinates Analysis (PCoA) of grassland communities under different fertilization regimes. Legend: T1—unfertilized control; T2—10 t ha^−1^ manure applied in autumn; T3—20 t ha^−1^ manure applied in autumn; T4—30 t ha^−1^ manure applied in autumn; T5—10 t ha^−1^ manure applied in spring; T6—20 t ha^−1^ manure applied in spring; T7—30 t ha^−1^ manure applied in spring; T8—Nitrocalc_20—200 kg ha^−1^ calcium ammonium nitrate applied in spring; T9—Nitrocalc_30—300 kg ha^−1^ calcium ammonium nitrate applied in spring; R1–R4—replications. Vectors indicate the fertilization gradient: *No_fertilz*—no fertilization (unfertilized control); *Fert_low_aut*—10 t ha^−1^ manure applied in autumn (T2); *Fert_med_aut*—20 t ha^−1^ manure applied in autumn (T3); *Fert_high_aut*—30 t ha^−1^ manure applied in autumn (T4); *Fert_low_spr*—10 t ha^−1^ manure applied in spring (T5); *Fert_med_spr*—20 t ha^−1^ manure applied in spring (T6); *Fert_high_spr*—30 t ha^−1^ manure applied in spring (T7); *Nitrocalc*_20—200 kg ha^−1^ nitrocalcar applied in spring (T8); and *Nitrocalc*_30–300 kg ha^−1^ nitrocalcar applied in spring (T9). Species abbreviations: Agrocapi—*Agrostis capillaris* L.; Festrubr—*Festuca rubra* L.; Nardstri—*Nardus stricta* L.; Avenplani—*Avenula planiculmis* (Schur) Holub; Anthodor—*Anthoxanthum odoratum* L.; Phlealpi—*Phleum alpinum* L.; Phleprat—*Phleum pratense* L.; Poachaix—*Poa chaixii* Vill.; Descflex—*Deschampsia flexuosa* (L.) Trin.; Trifhybr—*Trifolium hybridum* L.; Trifprat—*Trifolium pratense* L.; Trifrepe—*Trifolium repens* L.; Lotucorn—*Lotus corniculatus* L.; Achimill—*Achillea millefolium* L.; Achistri—*Achillea stricta* Schleich. ex W.D.J. Koch; Alchxant—*Alchemilla xanthochlora* Rothm.; Campserr—*Campanula serrata* (Kit.) Hendr.; Centphry—*Centaurea phrygia* L.; Crucglab—Cruciata glabra (L.) Ehrend.; Dianbarb—*Dianthus barbatus* subsp. compactus (Kit.) Grognot; Euphstri—*Euphrasia stricta* D. Wolff ex. J.F. Lehm.; Hieraura—*Hieracium aurantiacum* L.; Hierumbel—*Hieracium umbellatum* L.; Hypemacu—*Hypericum maculatum* Crantz; Hypounif—*Hypochaeris uniflora* Vill.; Leucvulg—*Leucanthemum vulgare* Lam.; Luzuluzu—*Luzula luzuloides* (Lam.) Dandy & Wilmott; Phytspic—*Phyteuma spicatum* L.; Poteerec—*Potentilla erecta* (L.) Raeusch.; Rhinmino—*Rhinanthus minor* L.; Rumeacet—*Rumex acetosa* L.; Scorrose—*Scorzonera rosea* Waldst. & Kit.; Stelgram—*Stellaria graminea* L.; Verocham—*Veronica chamaedrys* L.; Violcani—*Viola canina* L.; Violdecl—*Viola declinata* Waldst. & Kit.

**Figure 5 plants-15-00080-f005:**
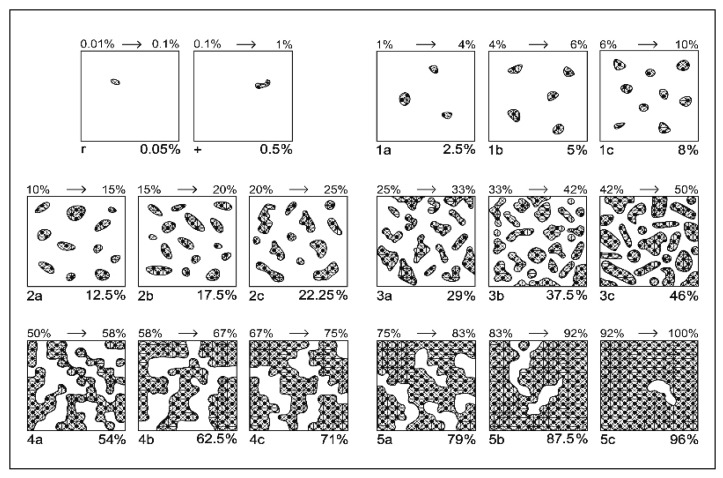
Modified Braun–Blanquét scale for grasslands based on species coverage [87]. Legend: 1 to 5 indicate the class of coverage; **a**, **b**, **c** indicate the sub-note of each class.

**Table 1 plants-15-00080-t001:** Importance of PCoA axis.

Axis	Degree of Participation (r)	Cumulative
1	0.921	0.921
2	0.067	0.988
3	0.001	0.989

Note: The correlations between the treatments and the PCoA axes (Table 2) showed that negative values on Axis 1 corresponded to unfertilized variants or variants with low organic input, and positive values on Axis 1 were associated with mineral fertilization and high doses of manure.

**Table 2 plants-15-00080-t002:** Correlation of experimental factors with ordination axes (PCoA).

Experimental Factors	Axis 1 (r)	Significance	Axis 2 (r)	Significance
No fertilization	−0.436	*	0.768	***
10t_man_aut	−0.340	*	−0.162	ns
20t_man_aut	−0.144	ns	−0.580	**
30t_man_aut	0.237	ns	−0.093	ns
10t_man_spri	−0.336	*	0.008	ns
20t_man_spri	−0.217	ns	−0.238	ns
30t_man_spri	0.213	ns	−0.033	ns
Nitrocalc_20	0.443	**	0.093	ns
Nitrocalc_30	0.582	**	0.238	ns
Axis importance	92.1%		6.7%	

Note: *r* = correlation coefficient between ordination distances and explanatory variables; significance: *** *p* < 0.001, ** *p* < 0.01, * *p* < 0.05, ns = not significant. Axis importance indicates the proportion of the total variance explained by each axis: Axis 1 = 92.1%, Axis 2 = 6.7%, cumulative = 98.8%.

**Table 3 plants-15-00080-t003:** MRPP pairwise comparisons of floristic composition between the four grassland groups.

Groups Compared	T Statistic	A (Within-Group Agreement)	*p*-Value
Group 1 vs. Group 2	−7.080	0.485	<0.001
Group 1 vs. Group 3	−4.450	0.721	<0.001
Group 1 vs. Group 4	−7.314	0.682	<0.001
Group 1 vs. Group 5	−4.444	0.609	<0.001
Group 1 vs. Group 6	−7.280	0.767	<0.001
Group 2 vs. Group 3	−6.736	0.458	<0.001
Group 2 vs. Group 4	−10.122	0.622	<0.001
Group 2 vs. Group 5	−6.282	0.284	<0.001
Group 2 vs. Group 6	−10.107	0.745	<0.001
Group 3 vs. Group 4	−7.253	0.562	<0.001
Group 3 vs. Group 5	−4.273	0.400	<0.001
Group 3 vs. Group 6	−7.191	0.719	<0.001
Group 4 vs. Group 5	−7.248	0.534	<0.001
Group 4 vs. Group 6	−9.513	0.462	<0.001
Group 5 vs. Group 6	−7.207	0.702	<0.001

Note: T = test statistics; A = chance-corrected within-group agreement. Group 1 = T1—unfertilized control; Group 2 = T2, T5 (T2—10 t ha^−1^ manure applied in autumn; T5—10 t ha^−1^ manure applied in spring); Group 3 = T6—20 t ha^−1^ manure applied in spring; Group 4 = T3—20 t ha^−1^ manure applied in autumn; Group 5 = T4, T7 (T4—30 t ha^−1^ manure applied in autumn; T7—30 t ha^−1^ manure applied in spring); Group 6 = T8, T9 (T8—200 kg ha^−1^ nitrocalcar applied in spring; T9—300 kg ha^−1^ nitrocalcar applied in spring).

**Table 4 plants-15-00080-t004:** Plant species correlation with ordination axes (PCoA).

Species	Axis 1 (r)	Axis 1 (r-sq)	Axis 1 (tau)	Signif.	Axis 2 (r)	Axis 2 (r-sq)	Axis 2 (tau)	Signif.
*Agrostis capillaris* L.	0.994	0.989	0.915	***	0.082	0.007	0.126	ns
*Festuca rubra* L.	−0.842	0.708	−0.559	***	−0.407	0.165	−0.548	*
*Nardus stricta* L.	−0.844	0.713	−0.801	***	0.491	0.241	0.035	**
*Avenula planiculmis* (Schur) Holub	−0.499	0.249	−0.534	**	0.445	0.198	−0.024	**
*Anthoxanthum odoratum* L.	−0.581	0.338	−0.403	**	−0.687	0.472	−0.592	***
*Phleum alpinum* L.	−0.532	0.283	−0.592	**	−0.146	0.021	−0.140	ns
*Phleum pratense* L.	0.842	0.709	0.773	***	−0.260	0.068	−0.026	ns
*Poa chaixii* Vill.	−0.512	0.262	−0.420	**	−0.253	0.064	−0.316	ns
*Deschampsia flexuosa* (L.) Trin.	−0.701	0.491	−0.584	***	0.154	0.024	−0.004	ns
*Trifolium hybridum* L.	−0.331	0.110	−0.310	*	−0.596	0.355	−0.537	**
*Trifolium pratense* L.	0.350	0.123	0.336	*	−0.670	0.449	−0.389	**
*Trifolium repens* L.	−0.346	0.120	−0.209	*	−0.768	0.590	−0.663	***
*Lotus corniculatus* L.	−0.488	0.238	−0.357	**	−0.647	0.419	−0.544	***
*Achillea millefolium* L.	−0.032	0.001	−0.209	ns	−0.157	0.025	−0.260	ns
*Achillea stricta* Schleich. ex W.D.J. Koch	−0.695	0.483	−0.531	***	−0.046	0.002	−0.155	ns
*Alchemilla xanthochlora* Rothm.	−0.422	0.178	−0.386	**	−0.262	0.069	−0.436	ns
*Campanula serrata* (Kit.) Hendrych	−0.312	0.097	−0.347	ns	−0.033	0.001	−0.202	ns
*Centaurea phrygia* L.	0.149	0.022	0.212	ns	−0.371	0.138	−0.379	*
*Cruciata glabra* (L.) Ehrend.	−0.426	0.182	−0.360	**	−0.512	0.262	−0.425	**
*Dianthus barbatus* subsp. *compactus* (Kit.) Grognot	−0.429	0.184	−0.468	**	−0.207	0.043	−0.362	ns
*Euphrasia stricta* D. Wolff ex J.F. Lehm.	−0.764	0.583	−0.636	***	0.093	0.009	−0.078	ns
*Hieracium aurantiacum* L.	−0.546	0.298	−0.611	**	0.133	0.018	−0.140	ns
*Hieracium umbellatum* L.	−0.579	0.336	−0.624	**	0.289	0.083	−0.083	ns
*Hypericum maculatum* Crantz	−0.365	0.133	−0.270	*	−0.591	0.349	−0.502	**
*Hypochaeris uniflora* Vill.	−0.688	0.473	−0.532	***	−0.205	0.042	−0.249	ns
*Leucanthemum vulgare* Lam.	−0.076	0.006	0.066	ns	−0.520	0.270	−0.306	**
*Luzula luzuloides* (Lam.) Dandy & Wilmott	−0.624	0.390	−0.551	***	0.336	0.113	0.159	*
*Phyteuma spicatum* L.	−0.311	0.097	−0.261	ns	−0.232	0.054	−0.270	ns
*Potentilla erecta* (L.) Raeusch.	−0.774	0.599	−0.753	***	0.366	0.134	−0.005	*
*Rhinanthus minor* L.	−0.614	0.377	−0.582	***	0.081	0.007	−0.065	ns
*Rumex acetosa* L.	−0.526	0.277	−0.399	**	−0.062	0.004	−0.160	ns
*Scorzonera rosea* Waldst. & Kit.	−0.879	0.772	−0.672	***	−0.171	0.029	−0.327	ns
*Stellaria graminea* L.	−0.554	0.306	−0.387	**	−0.436	0.190	−0.449	*
*Veronica chamaedrys* L.	−0.334	0.111	−0.257	*	−0.628	0.394	−0.619	***
*Viola canina* L.	−0.641	0.411	−0.593	***	0.453	0.205	0.242	**
*Viola declinata* Waldst. & Kit.	−0.679	0.461	−0.546	***	0.137	0.019	−0.012	ns

Note: *r*— Pearson’s correlation coefficient; r-sq—coefficient of determination; tau—Kendall’s tau correlation between ordination scores and species abundance. Significance: *** *p* < 0.001, ** *p* < 0.01, * *p* < 0.05, ns = not significant.

**Table 5 plants-15-00080-t005:** Indicator value of species related to the groups.

Species	Group	IndVal	Signif.
*Agrostis capillaris* L.	6	46.4	*p* < 0.001
*Festuca rubra* L.	3	22.1	ns
*Nardus stricta* L.	1	41.9	*p* < 0.001
*Avenula planiculmis* (Schur) Holub	1	50.0	*p* < 0.01
*Anthoxanthum odoratum* L.	3	39.5	*p* < 0.01
*Phleum alpinum* L.	2	51.1	*p* < 0.01
*Phleum pratense* L.	6	30.4	*p* < 0.01
*Poa chaixii* Vill.	2	40.9	*p* < 0.01
*Deschampsia flexuosa* (L.) Trin.	1	26.7	ns
*Trifolium hybridum* L.	3	48.1	*p* < 0.01
*Trifolium pratense* L.	4	23.3	*p* < 0.05
*Trifolium repens* L.	3	31.0	*p* < 0.01
*Lotus corniculatus* L.	3	40.3	*p* < 0.01
*Achillea millefolium* L.	4	34.7	ns
*Achillea stricta* Schleich. ex W.D.J. Koch	1	29.6	ns
*Alchemilla xanthochlora* Rothm.	5	47.1	*p* < 0.01
*Campanula serrata* (Kit.) Hendrych	5	27.0	ns
*Centaurea phrygia* L.	4	39.2	*p* < 0.01
*Cruciata glabra* (L.) Ehrend.	3	46.5	*p* < 0.01
*Dianthus barbatus* subsp. *compactus* (Kit.) Grognot	5	40.0	*p* < 0.05
*Euphrasia stricta* D. Wolff ex J.F. Lehm.	1	30.8	ns
*Hieracium aurantiacum* L.	1	34.8	ns
*Hieracium umbellatum* L.	1	38.2	ns
*Hypericum maculatum* Crantz	3	39.5	*p* < 0.01
*Hypochaeris uniflora* Vill.	2	26.1	*p* < 0.05
*Leucanthemum vulgare* Lam.	3	49.0	*p* < 0.01
*Luzula luzuloides* (Lam.) Dandy & Wilmott	1	40.0	*p* < 0.05
*Phyteuma spicatum* L.	5	30.0	ns
*Potentilla erecta* (L.) Raeusch.	1	36.4	*p* < 0.05
*Rhinanthus minor* L.	2	50.0	*p* < 0.01
*Rumex acetosa* L.	1	20.0	ns
*Scorzonera rosea* Waldst. & Kit.	1	23.5	ns
*Stellaria graminea* L.	2	22.9	ns
*Veronica chamaedrys* L.	3	34.8	*p* < 0.05
*Viola canina* L.	1	53.3	*p* < 0.01
*Viola declinata* Waldst. & Kit.	1	22.5	ns

Note: Full results of the Indicator Species Analysis (ISA) showing indicator values (IndVal) and statistical significance (*p*-values) for all species across the six floristic groups identified by cluster analysis. Significance: ns = not significant. Groups meaning: Group 1 = T1—unfertilized control; Group 2 = T2, T5 (T2—10 t ha^−1^ manure applied in autumn; T5—10 t ha^−1^ manure applied in spring); Group 3 = T6—20 t ha^−1^ manure applied in spring; Group 4 = T3—20 t ha^−1^ manure applied in autumn; Group 5 = T4, T7 (T4—30 t ha^−1^ manure applied in autumn; T7—30 t ha^−1^ manure applied in spring); Group 6 = T8, T9 (T8—Nitrocalc_20—200 kg ha^−1^ calcium ammonium nitrate applied in spring; T9—Nitrocalc_30—300 kg ha^−1^ calcium ammonium nitrate applied in spring).

**Table 6 plants-15-00080-t006:** The influence of fertilization treatments on biodiversity indices in grasslands.

Treatment	Species Richness (S)	Shannon Index (H’)	Evenness (E)	Simpson (D)
T1	31.75 ± 0.96	2.199 ± 0.055	0.636 ± 0.016	0.789 ± 0.018
T2	32.50 ± 1.73	2.714 ± 0.043	0.780 ± 0.011	0.892 ± 0.006
T3	32.25 ± 1.50	2.770 ± 0.035	0.798 ± 0.004	0.903 ± 0.003
T4	26.25 ± 2.50	2.158 ± 0.036	0.662 ± 0.025	0.779 ± 0.007
T5	35.00 ± 1.15	2.857 ± 0.042	0.804 ± 0.016	0.912 ± 0.004
T6	32.25 ± 2.22	2.709 ± 0.037	0.780 ± 0.011	0.895 ± 0.003
T7	24.25 ± 1.71	1.958 ± 0.070	0.615 ± 0.021	0.744 ± 0.017
T8	18.25 ± 5.12	1.426 ± 0.085	0.500 ± 0.032	0.548 ± 0.009
T9	11.25 ± 2.22	0.926 ± 0.027	0.388 ± 0.037	0.387 ± 0.017
F-test	43.05 (df = 8.27)	685.81 (df = 8.27)	183.73 (df = 8.27)	1079.41 (df = 8.27)
*p*-value	<0.001	<0.001	<0.001	<0.001

Legend: T1—unfertilized control; T2—10 t ha^−1^ manure applied in autumn; T3—20 t ha^−1^ manure applied in autumn; T4—30 t ha^−1^ manure applied in autumn; T5—10 t ha^−1^ manure applied in spring; T6—20 t ha^−1^ manure applied in spring; T7—30 t ha^−1^ manure applied in spring; T8—Nitrocalc_20—200 kg ha^−1^ calcium ammonium nitrate applied in spring; T9—Nitrocalc_30—300 kg ha^−1^ calcium ammonium nitrate applied in spring. Note: Values are means ± standard error (n = four replicates per treatment). One-way ANOVA was used to test for differences between the treatments.

**Table 7 plants-15-00080-t007:** Modified Braun–Blanquét scale for assessing the abundance–dominance of plant species based on classes and sub-classes [87].

Class	Coverage Interval (%)	Class Central Value (%)	Sub-Note	Sub-Interval (%)	Central-Adjusted Value of Sub-Interval (%)
5	75–100	87.5	5 c	92–100	96
			5 b	83–92	87.5
			5 a	75–83	79
4	50–75	62.5	4 c	67–75	71
			4 b	58–67	62.5
			4 a	50–58	54
3	25–50	37.5	3 c	42–50	46
			3 b	33–42	37.5
			3 a	25–33	29
2	10–25	17.5	2 c	20–25	22.25
			2 b	15–20	17.5
			2 a	10–15	12.5
1	1–10	5	1 c	6–10	8
			1 b	4–6	5
			1 a	1–4	2.5
+	0.1–1	0.5	-	-	0.5
r	0.01–0.1	0.05	-	-	0.05

Note: a, b, and c indicate the sub-notes of each class.

## Data Availability

No new data were created or analyzed in this study.

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
