# Peer review of "The Effect of Fertilization on Floristic Composition and Biodiversity of Montane Grasslands (HNV) in the Eastern Carpathians"

_plants, 2025, doi:10.3390/plants15010080_

Round 1

Reviewer 1 Report

Comments and Suggestions for Authors

Fertilization is one of the key measures in grassland management, with most mountain grasslands showing high sensitivity to fertilization intensity. Therefore, systematically evaluating fertilization practices is important for research on grassland soil-plant systems. This paper, based on a three-year monitoring experiment, found that moderate organic fertilizer management best promotes the synergy between biodiversity and ecosystem functions. This finding is consistent with prevailing international research perspectives and provides scientific guidance for the sustainable management of mountain grasslands worldwide. While this paper has certain strengths, it currently remains largely descriptive, lacking sufficient depth in exploring the underlying ecological mechanisms (such as the driving mechanisms of soil microorganisms). Specific comments are as follows:

  1. The introduction mainly lists literature without in-depth commentary. It is recommended that the authors begin by addressing the context of biodiversity loss under global change, focusing on anthropogenic disturbance factors, and conduct a thorough review.
  2. The methodology section focuses on vegetation responses, with limited detail on how fertilization treatments specifically alter soil biological and physicochemical properties. If such data exist, their inclusion is recommended.
  3. The study area has a research history spanning over two decades. Readers may wish to observe the long-term trends or evolutionary trajectory of the grassland community from 2002 to the present. It is recommended that the authors briefly describe these long-term trends in the study area overview or discussion section.
  4. In the results section, the authors should focus solely on presenting the findings from this specific experiment, avoiding overlap with the discussion section.
  5. The conclusion that ‘moderate organic fertilizer application is beneficial’ aligns with prevailing international perspectives. However, the study's innovation should be demonstrated more clearly through specific action mechanisms and long-term succession trajectories. It is recommended to incorporate relevant models to elucidate plant competition mechanisms, such as niche theory and light competition models. Under high-intensity fertilization, why do certain plants become dominant species? What mechanisms underlie this phenomenon?
  6. Overall, this paper focuses predominantly on aboveground vegetation systems and lacks critical soil indicator data. Supplementing soil nutrient data would establish a comprehensive framework linking fertilization practices, soil environment, and plant communities, significantly enhancing the paper's quality.
  7. The authors should explicitly state the limitations of this study in the concluding remarks, particularly regarding the driving mechanisms within soil ecology.
  8. The authors are advised to review the entire manuscript to ensure adherence to basic writing conventions, such as presenting all p-values in italics.

Author Response

We thank the reviewer for their suggestions and comments, to which we respond below:

Comment 1: The introduction mainly lists the literature, without in-depth commentary. It is recommended that the authors begin by addressing the context of biodiversity loss in the context of global change, focusing on anthropogenic disturbance factors, and conduct an in-depth analysis.

Response 1: We thank the reviewer for this constructive and pertinent comment. In response, we have revised the introduction in detail to provide a clearer and well-documented conceptual framework.

Comment 2: The methodology section focuses on vegetation responses, with limited details on how fertilization treatments specifically alter the biological and physicochemical properties of the soil. If such data exist, their inclusion is recommended.

Response 2: We thank the reviewer for this observation. We fully agree that the effects of fertilization on vegetation are mediated by changes in the physicochemical and biological processes of the soil. In the present study, however, the experimental design and analytical framework intentionally focused only on vegetation-based indicators (floristic composition, diversity indices, ordination, and indicator species) as early integrative responses of the soil-plant system to contrasting fertilization regimes.

To clarify this methodological choice, we added a note at the end of the Materials and Methods section (Section 4.4, Data Analysis), indicating that the analysis of changes in the physicochemical properties of the soil will be carried out in the next stages of this long-term experiment.

Comment 3: The study area has a research history spanning more than two decades. Readers may wish to observe the long-term trends or evolutionary trajectory of the grassland community from 2002 to the present. It is recommended that the authors briefly describe these long-term trends in the study area overview section or in the discussion section.

Response 3: We thank the reviewer for this suggestion. The issues raised are clarified by correcting the error regarding the start date of the current experiment. To address this comment, we have added a concise paragraph to the Discussion section (section 3.1), summarizing previously published long-term conclusions from the Eastern and Western Carpathians and adjacent regions.

At the same time, we explicitly clarify that the experimental plots analyzed in the present study were established in 2021 and not 2002 as erroneously recorded in the original manuscript. The newly added text contextualizes our results within a broader regional and historical framework, based on previous studies by different authors.

Comment 4: In the results section, the authors should focus exclusively on presenting the conclusions of this specific experiment, avoiding overlap with the discussion section.

Response 4: We thank the reviewer for this constructive comment.

In response, we have carefully revised the Results section to ensure that it presents only the empirical conclusions of the present experiment. All interpretive statements, references to ecological mechanisms, comparisons with the literature, and management implications have been removed from the Results section and moved to the appropriate subsections of the Discussion section.

Comment 5: The conclusion that "moderate application of organic fertilizers is beneficial" is consistent with prevailing international perspectives. However, the innovative nature of the study should be more clearly demonstrated through specific mechanisms of action and long-term succession trajectories. It is recommended to incorporate models relevant to elucidating plant competition mechanisms, such as niche theory and light competition models. Under conditions of intensive fertilization, why do certain plants become dominant species? What mechanisms underlie this phenomenon?

Response 5: We thank the reviewer for this pertinent comment.

In response, we have strengthened the Discussion section by explicitly linking the observed vegetation patterns to well-established ecological mechanisms and theoretical frameworks, including niche differentiation, competitive exclusion, and competition for light. We now explain that, under conditions of intensive fertilization, increased nutrient availability—particularly nitrogen—favors rapid biomass accumulation by fast-growing grasses (e.g., Agrostis capillaris), leading to asymmetric competition for light. This results in shading, suppression of slow-growing oligotrophic species, and a rapid decline in functional and taxonomic diversity, consistent with the classical theory of competition and resource ratio. In contrast, moderate organic fertilization maintains intermediate nutrient availability and structural heterogeneity, which relaxes competitive dominance and allows the coexistence of species with contrasting ecological strategies. This mechanism is discussed in the context of the intermediate disturbance hypothesis and niche-based coexistence models, providing a process-based explanation for the maximum diversity observed under moderate manure input conditions.

Comment 6: Overall, this article focuses primarily on surface vegetation systems and does not contain critical data on soil indicators. The addition of soil nutrient data would create a comprehensive framework linking fertilization practices, soil environment, and plant communities, significantly improving the quality of the article.

Response 6: We thank the reviewer for this observation.

In the present study, our main objective was to evaluate floristic composition, vegetation typology, and biodiversity responses to contrasting fertilization regimes using a standardized experimental design that will be followed up in the long term. As such, the methodological focus of this manuscript was on assessing the dynamics of the surface vegetation community, and changes at the soil level will be analyzed after a longer period of experimentation and will be the subject of further analysis, as specified in the Conclusions section, in the paragraph Limitations of the study and future research directions.

Comment 7: The authors should explicitly mention the limitations of this study in the conclusions, especially with regard to the determining mechanisms in soil ecology.

Response 7: We thank the reviewer for this pertinent suggestion.

In response, we have revised the Conclusions section by introducing the issues mentioned as limitations of the present study and directions for future research.

Comment 8: The authors are advised to review the entire manuscript to ensure that it complies with basic writing conventions, such as presenting all p-values in italics.

Response 8: We thank the reviewer for this editorial comment. In response, we have carefully reviewed the entire manuscript to ensure that it fully complies with standard scientific writing conventions and the formatting guidelines of the journal Plants.

We are confident that these revisions will improve the clarity, accuracy, and professional presentation of our manuscript, and we thank the reviewer once again for their comments and suggestions.

Reviewer 2 Report

Comments and Suggestions for Authors

General Comments:

This manuscript addresses a highly relevant topic regarding the conservation and sustainable management of High Nature Value (HNV) grasslands in the Eastern Carpathians. The study utilizes a so-called long-term experimental setup to assess the impact of different organic and mineral fertilization regimes on floristic composition and diversity. The overall methodology is sound, employing appropriate multivariate statistical tools (clustering, PCoA, MRPP, ISA). The findings, particularly the adherence to the Intermediate Disturbance Hypothesis (IDH) with moderate organic fertilization, provide valuable, actionable management recommendations for policymakers and land managers. The paper is well-structured and the conclusions are supported by the presented data.

However, several major issues related to data scope and presentation need to be addressed before acceptance.

Major Comments  - what must be improved:

The Abstract and Introduction clearly state that the study is based on a long-term experiment established in 2002 but presents results from three recent experimental years (2022–2024)1. This creates a significant gap between the described long-term experimental framework and the short-term data presented. To genuinely demonstrate the "effect" of a long-term experiment, the authors must justify this selection more thoroughly or, ideally, include data from earlier key years (e.g., initial state in 2002, or a mid-point year). Without this, the results primarily reflect the short-term state of vegetation under management that has been applied over two decades, not the trajectory of change. The current data selection undermines the novelty of using a long-term plot network. The self-acknowledged limitation (Section 3.5, lines 476–479) 2 is insufficient; this is a presentation issue that needs fixing.

The methods section lists Multi-Response Permutation Procedure (MRPP) and Indicator Species Analysis (ISA) as key analytical tools. But the results for MRPP (Table 3 in the original manuscript, partially visible in snippets) confirm that all floristic groups are significantly different (p < 0.001). This is a good result, but the full table should be clearly presented, and its ecological meaning discussed more explicitly beyond simple statistical significance.

The Indicator Species Analysis (ISA) is critical for characterizing the ecologically defined clusters. The review is currently missing the full results table (Indicator Value, p-value, and specific indicator species) for all six floristic types. This table is essential for readers to understand which species drive the separation of the groups, especially the transition from Nardus stricta (oligotrophic) to Agrostis capillaris (eutrophic). So, the full ISA table must be included in the Results section.

Axis 1 of the PCoA explains an unusually high proportion of the total variation in species composition (92.1%). While this strongly supports the authors' claim of a clear, overriding trophic gradient, it is worth a brief discussion or a stronger statement in the Discussion section to contextualize this. The authors should explicitly comment on what this extreme dominance of Axis 1 implies for the grassland system (e.g., that N-input is the single, overwhelmingly dominant ecological filter compared to other potential drivers like microtopography or soil chemistry heterogeneity).

 Minor Comments:

In the Abstract and Introduction, ensure consistency when listing the fertilization doses. The use of "Nitrocalcar" is stated in kg/ha, while manure is in t/ha. This is standard, but the exact composition of 'nitrocalcar' (an NPK or just N?) should be specified, at least as a footnote in the Introduction or Methods, as it’s a generic term.

The legend for Figure 1 uses the term "300 kg ha-1 nitrocalcar applied in spring" for T9. In the PCoA correlation table (Table 2), T9 is labelled as "Nitrocalcar 300 kg ha-1," but the associated text also refers to it as "Nitrocalc_30”. Similarly, the PCoA figure vectors use "Nitrocalc_20" and "Nitrocalc_30”. For consistency, ensure all references (text, tables, and figures) use the same treatment abbreviation/labeling style.

Comments on the Quality of English Language

The overall language is good, but some phrasing could be refined for flow:

Line 478 states: "Given the short duration (3 years) of the experiment...”. Change "of the experiment" to "of the presented dataset" for greater precision, as the experiment itself is long-term.

In the PCoA section, line 166 states "Mineral treatments (T8–T9) and some variants with high doses of manure (T4 and T7) were applied in areas dominated by Agrostis capillaris". Reword to: "...were located/grouped in the ordination space associated with Agrostis capillaris dominance."

Author Response

Thank you for your appreciation of the relevance of our study, as well as for your suggestions and comments, to which we respond below:

Comment 1: The abstract and introduction clearly state that the study is based on a long-term experiment initiated in 2002, but present results from three recent experimental years (2022-2024). This creates a significant discrepancy between the long-term experimental framework described and the short-term data presented.

To authentically demonstrate the "effect" of a long-term experiment, the authors need to justify this selection in more detail or, ideally, include data from previous key years (e.g., the initial state in 2002 or an intermediate year). Without this, the results primarily reflect the short-term state of vegetation under management that has been applied over two decades, not the trajectory of change. The current selection of data undermines the novelty of using a long-term plot network. The acknowledged limitation (section 3.5, lines 476-479) 2 is insufficient; this is a presentation issue that needs to be addressed.

Response 1: We thank the reviewer for pointing out this inconsistency regarding the duration of the experiment. We note that the fertilization experiment was initiated in 2021, not 2002, as erroneously stated in the previous version of the manuscript. We regret this typographical error. References to this issue in the abstract and introduction have been corrected in the current version of the manuscript, and section 3.5 has also been improved. Consequently, our study presents results from the period 2022-2024, so it is not necessary to reconstruct long-term succession trajectories.

Comment 2: The methods section lists the multiple response permutation procedure (MRPP) and indicator species analysis (ISA) as key analytical tools. But the results for MRPP (Table 3 in the original manuscript, partially visible in excerpts) confirm that all floristic groups are significantly different (p < 0.001). This is a good result, but the complete table should be clearly presented, and its ecological significance should be discussed more explicitly, beyond mere statistical significance.

Response 2: We thank the reviewer for this comment. In the revised manuscript, the complete table of MRPP pairwise comparisons is included in the Results section (Table 3), detailing the T-statistic, A-value (group-internal concordance), and associated p-values for all floristic group comparisons.

In addition, we have expanded the Results and Discussion sections to explicitly interpret the ecological significance of the MRPP results, emphasizing that consistently high A values indicate strong homogeneity within the group and well-defined floristic types along the fertilization gradient.

Comment 3: Indicator species analysis (ISA) is essential for characterizing ecologically defined groups. Currently, the review does not contain the complete table of results (indicator value, p-value, and specific indicator species) for all six floristic types.

This table is essential for readers to understand which species determine the separation of groups, particularly the transition from Nardus stricta (oligotrophic) to Agrostis capillaris (eutrophic). Therefore, the complete ISA table should be included in the Results section.

Response 3: We thank the reviewer for this constructive comment. In response, the complete table with the results of the indicator species analysis (ISA) is included in the Results section (Table 5). The table now explicitly presents, for all six floristic groups: (i) indicator species, (ii) corresponding indicator values (IndVal), and (iii) statistical significance (p-values), as requested.

In the revised manuscript, we have expanded the analysis of the results to clearly describe the ecological significance of the identified indicator species, highlighting in particular the transition from oligotrophic grasslands dominated by Nardus stricta to eutrophic communities dominated by Agrostis capillaris under conditions of increased fertilization intensity. This allows readers to make a direct link between species-level responses and the fertilization gradient and the floristic groups identified by hierarchical classification and PCoA analyses.

Comment 4: Axis 1 of the PCoA explains an unusually large proportion of the total variation in species composition (92.1%). Although this strongly supports the authors' claim of a clear and predominant trophic gradient, it warrants a brief discussion or stronger statement in the Discussion section to contextualize this. The authors should explicitly comment on what this extreme dominance of axis 1 implies for the grassland system (e.g., that nitrogen input is the only dominant ecological filter compared to other potential factors such as microtopography or soil chemical heterogeneity).

Response 4: We thank the reviewer for this suggestion.

In the revised manuscript, we have expanded the Discussion section to clarify that this result indicates the dominance of fertilization—specifically nitrogen input—as the main ecological filter structuring plant community composition in the studied grasslands.

Minor comments:

Comment 5: In the abstract and introduction, ensure consistency when listing fertilization doses. The use of "Nitrocalcar" is indicated in kg/ha, while manure is in t/ha. This is standard practice, but the exact composition of "nitrocalcar" (an NPK or just N?) should be specified, at least as a footnote in the introduction or methods, as it is a generic term.

Response 5: We thank the reviewer for pointing out this issue.

In the revised manuscript, we have specified the exact composition of the mineral fertilizer used. This clarification has been added to the Materials and Methods section (fertilization treatments) and mentioned consistently throughout the revised manuscript.

Comment 6: The legend for Figure 1 uses the term "300 kg ha-1 nitrocalcar applied in spring" for T9. In the PCoA correlation table (Table 2), T9 is labeled as "Nitrocalcar 300 kg ha-1," but the associated text also refers to it as "Nitrocalc_30." Similarly, the vectors in the PCoA figure use "Nitrocalc_20" and "Nitrocalc_30." For consistency, please ensure that all references (text, tables, and figures) use the same abbreviation/labeling style for the treatment.

Response 6: We thank the reviewer for this comment. In the revised manuscript, we have standardized the terminology throughout the text, tables, and figures. All mineral fertilizer treatments are now consistently labeled as Nitrocalc_20 and Nitrocalc_30, corresponding to 200 and 300 kg ha⁻¹ of calcium and ammonium nitrate applied in the spring. The same abbreviations are used consistently in Figure 1, Figure 2 (PCoA), Table 2, and in the Results and Discussion sections.

Comments on the quality of the English language

Overall, the language is good, but some wording could be improved to ensure the fluidity of the text:

Comment 7: Line 478 states: "Given the short duration (3 years) of the experiment...". Change "of the experiment" to "of the presented dataset" for greater accuracy, as the experiment itself is long-term.

Response 7: We thank the reviewer for the suggestion. We have revised the sentence by replacing "the short duration of the experiment" with "the short duration of the presented dataset."

Comment 8: In the PCoA section, line 166 states, "Mineral treatments (T8–T9) and some high manure treatments (T4 and T7) were applied in areas dominated by Agrostis capillaris." Reword as follows: "...were located/grouped in the ordination space associated with Agrostis capillaris dominance."

Response 8: We thank the reviewer for the suggestion. We have revised the wording as requested.

Reviewer 3 Report

Comments and Suggestions for Authors

The overall quality of the paper is relatively high. It can be accepted for publication after minor revisions. Specific details for modification are as follows:

Line 62: Although the purpose of this sentence is to highlight the role of mineral fertilization, the description regarding "Mulching" maintaining HNV grasslands is not clearly articulated. Therefore, the brief mention of "Mulching" can actually be deleted, unless you intend to describe "Mulching" in detail and use it as a control in subsequent research.

Line 76: Latin names of plants need to be in italics.

Line 490: Remove the period "." in the title "Study area and pedoclimatic conditions."

Lines 496-502: This sentence is too long and contains grammatical errors. Please break it into shorter sentences.

Tables 7 to 10: Please present them using the three-line table format. The meaning of some column headers in the tables is unclear; please modify them or provide explanations in appropriate locations. In the table data, what does the comma in numbers like "6,5" signify? Is it a decimal point? Please verify and correct accordingly. These tables are not the main data of the paper and can be included as supplementary tables.

Line 581: "Floristic-ecological classification." This is an incomplete sentence. Please check and confirm.

Line 144: "The strongest link was between Nitrocalc_30 and Axis 1" – It needs to be specified what value of Axis 1 is being referred to, such as its eigenvalue.

Line 273: The title "2.4. Species–environment relationships along PCoA axes" does not match the content. The title suggests relationships between species and the environment, but the content discusses relationships between species abundance and ordination axis scores. Please revise accordingly.

In Tables 4 and 5, which are in three-line format, one line is missing.

In Table 6, "Species richnes" in the F-test value is missing "df=827".

Line 412: Please check if "6230* habitats" is correct.

It is recommended to integrate the sections "3.5. Study Limitations" and "3.6. Future Research Directions" into the "Conclusion" section.

Author Response

Thank you for your appreciation of the overall quality of our article, as well as for your suggestions and comments, to which we respond below:

Comment 1: Line 62: Although the purpose of this sentence is to highlight the role of mineral fertilization, the description of "mulching" that maintains high natural value grasslands is not clearly articulated. Therefore, the brief mention of "mulching" can actually be removed, unless you intend to describe "mulching" in detail and use it as a control element in further research.

 Response 1: We thank the reviewer for this observation. We have removed the reference to mulching from the introduction and revised the wording in the manuscript. This revision ensures that all management practices discussed in the Introduction are addressed and evaluated directly in the study.

Comment 2: Line 76: Latin plant names should be written in italics.

 Response 2: We thank the reviewer for pointing out this formatting issue. We have checked and implemented the requirement that all Latin names of plant species be italicized throughout the manuscript, in accordance with botanical nomenclature standards and the journal's style requirements.

Comment 3: Line 490: Remove the period "." from the title "Study area and pedoclimatic conditions."

 Response 3: We thank the reviewer for this observation. The request has been addressed.

Comment 4: Lines 496-502: This sentence is too long and contains grammatical errors. Please split it into shorter sentences.

Response 4: We thank the reviewer for this suggestion. We have revised the sentence and corrected the grammatical errors.

Comment 5: Tables 7-10: Please present them using the three-line table format. The meaning of some column headers in the tables is unclear; please modify them or provide explanations in the appropriate places. In the data in the table, what is the meaning of the comma in numbers such as "6.5"? Is it a decimal point?

Please check and correct accordingly. These tables do not represent the main data of the article and can be included as supplementary tables.

 Response 5: We thank the reviewer for their comments. Tables 7-10 have been revised accordingly, checked, and all numerical values have been corrected in accordance with international scientific standards and moved to the Supplementary Materials section as suggested, and the corresponding references to these tables have been retained in the main text.

Comment 6: Line 581: "Floristic-ecological classification." This is an incomplete sentence. Please check and confirm.

 Response 6: We thank the reviewer for pointing out this issue. We have corrected the wording in the current version of the manuscript.

Comment 7: Line 144: "The strongest link was between Nitrocalc_30 and Axis 1" – The value of Axis 1 referred to, such as the eigenvalue, should be specified.

Response 7: We thank the reviewer for this observation. The text has been revised to explicitly specify the value associated with Axis 1. This clarification improves the accuracy of the interpretation and avoids ambiguity.

Comment 8: Line 273: The title "2.4. Relationships between species and environment along PCoA axes" does not correspond to the content. The title suggests relationships between species and environment, but the content discusses relationships between species abundance and ordination axis scores. Please revise accordingly.

 Response 8: We thank the reviewer for the suggestion. The title has been revised.

Comment 9: In Tables 4 and 5, which are in a three-line format, one line is missing.

Response 9: We thank the reviewer for the observation. Tables 4 and 5 have been revised in accordance with the standard format recommended by the journal.

Comment 10: In Table 6, "Species rich" in the F-test value is missing "df=827".

Response 10: We thank the reviewer for pointing out this omission. The degrees of freedom for the F-test for species richness in Table 6 have now been added.

Comment 11: Row 412: Please check if "6230* habitats" is correct.

 Response 11: We thank the reviewer for the suggestion. The name "6230* habitats" has been checked with and is correct, referring to the priority habitat type "Species-rich grasslands on siliceous substrates in mountain areas" according to Annex I of the EU Habitats Directive.

Comment 12: It is recommended that sections "3.5. Limitations of the study" and "3.6. Future research directions" be integrated into the "Conclusions" section.

Response 12: We appreciate this constructive suggestion. We have integrated sections "3.5. Limitations of the study" and "3.6. Future research directions" into the Conclusions section.

Round 2

Reviewer 1 Report

Comments and Suggestions for Authors

The author has made the necessary revisions, and the current version can be published.

Reviewer 2 Report

Comments and Suggestions for Authors

Dear Aurhors, 

Thank you for considering my suggestions. Now your work sounds better!